# Sparse CLIP: Co-Optimizing Interpretability and Performance in Contrastive Learning

**Chuan Qin[1]**    **Constantin Venhoff[2]***    **Sonia Joseph[1]**    **Fanyi Xiao[1]**    **Stefan Scherer[1]**

[1]Meta    [2]University of Oxford

## Abstract

Contrastive Language-Image Pre-training (CLIP) has become a cornerstone in vision-language representation learning, powering diverse downstream tasks and serving as the default vision backbone in multimodal large language models (MLLMs). Despite its success, CLIP's dense and opaque latent representations pose significant interpretability challenges. A common assumption is that interpretability and performance are in tension: enforcing sparsity during training degrades accuracy, motivating recent post-hoc approaches such as Sparse Autoencoders (SAEs). However, these post-hoc approaches often suffer from degraded downstream performance and loss of CLIP's inherent multimodal capabilities, with most learned features remaining unimodal.

We propose a simple yet effective approach that integrates sparsity directly into CLIP training, yielding representations that are both interpretable and performant. Compared to SAEs, our Sparse CLIP representations preserve strong downstream task performance, achieve superior interpretability, and retain multimodal capabilities. We show that multimodal sparse features enable straightforward semantic concept alignment and reveal training dynamics of how cross-modal knowledge emerges. Finally, as a proof of concept, we train a vision-language model on sparse CLIP representations that enables interpretable, vision-based steering capabilities. Our findings challenge conventional wisdom that interpretability requires sacrificing accuracy and demonstrate that interpretability and performance can be co-optimized, offering a promising design principle for future models.

## 1 Introduction

Contrastive Language-Image Pre-training (CLIP) (Radford et al., 2021) has achieved significant success in vision-language representation learning by linking images and text through contrastive training on web-crawled image-text pairs. Its transferability and generalizability have been demonstrated across a wide range of downstream tasks, including but not limited to zero-shot image classification and image-text retrieval. Even beyond, CLIP has arguably become the default choice for the vision backbone in multimodal large language models (MLLMs). Given the heavy reliance of these applications on the prior knowledge in CLIP embeddings, it is essential to improve our understanding to the representations that CLIP generated, which is a dense latent space not directly interpretable.

Recently, researchers have begun applying Sparse Autoencoders (SAEs)—a well-established tool for mechanistically interpreting large language models (LLMs)—to CLIP vision encoders, aiming to gain deeper insights into the concepts captured by pretrained CLIP models (Cunningham et al., 2023; Joseph et al., 2025b;a). Typically, an SAE introduces a bottleneck layer into the residual stream of a pretrained CLIP encoder. By enforcing sparsity during the training of this high-dimensional bottleneck, the SAE disentangles the original CLIP features into sparse representations, which can potentially be associated with specific concepts.

While SAEs enhance interpretability, several limitations constrain their practical utility. First, researches on LLM has shown that sparse SAE features often underperform on downstream tasks compared to their dense counterparts. For example, Kantamneni et al. (2025) shows that for probing

---

*Contribution made during internship at Meta.

task, SAE latents fail to achieve a consistent overall advantage over dense counterparts, and Farrell et al. (2024) has had similar result on a different unlearning task. Second, they lose the multi-modal capabilities of CLIP. Most CLIP SAEs were trained only on the residual stream of vision tower to get interpretable vision features. And when Papadimitriou et al. (2025) trained SAEs on the multimodal representation space of pretrained CLIP models, they found that most features were unimodal, activated exclusively for image or text inputs.

These limitations reflect a broader conventional view currently in the field interpretability: that interpretability and accuracy are fundamentally in tension. In particular, enforcing sparsity during training is widely assumed to degrade downstream task performance, which has motivated the focus on post-hoc sparse methods (Olshausen & Field, 1996; Higgins et al., 2017; Koh et al., 2020; Hendrycks & Hiscott, 2025). Although some work suggests that this trade-off may not be inevitable (Rudin, 2019), the prevailing assumption has remained that training-time sparsity compromises accuracy.

In this work, we challenge this assumption. We demonstrate that by projecting dense features into a high-dimensional space and enforcing sparsity during CLIP training, we can transform dense CLIP representations into interpretable features similar to those produced by SAEs. However, unlike SAE features, these "Sparse CLIP features" maintain strong performance on downstream tasks, and naturally preserve CLIP's multi-modal nature, as they are directly trained using a cross-modal cosine similarity loss. Furthermore, we show that this inherent multi-modal nature enables easy alignment of features with concepts. Building on this, we analyzed these multi-modal representations, revealing some intriguing multi-modal concepts that the model is able to learn, and more interestingly, how they emerge and evolve during training. Finally, as downstream task for applications, we trained a VLM using the sparse CLIP representations, and demonstrated vision-based steering using interpretable features.

We summarize our contributions as follows:

- We introduce a novel but simple technique for learning sparse CLIP representations that maintain competitive performance while achieving interpretable multimodal features.

- We demonstrate how multimodal alignment unlocks transparent analysis of CLIP's encoded knowledge, revealing feature evolution dynamics throughout training.

- We demonstrate real-world application of our approach by implementing interpretable vision-based steering in vision-language models.

## 2 SPARSE CLIP TRAINING

### 2.1 ADDING SPARSITY TO CLIP PRETRAINING

Taking a broader perspective, there exists a family of concept extraction methods designed to improve the interpretability of machine learning models. Fel et al. (2023) proposed that all concept extraction methods can be unified within the framework of dictionary learning. To faithfully interpret an activation matrix $A \in \mathbb{R}^{n \times m}$, where $n$ is the size of dataset and $m$ is the width of the activation, the goal is to find two low-rank matrices $(U, V)$ such that $A \approx UV^T$. In this framework, $V \in \mathbb{R}^{m \times k}$ serves as the dictionary while $U \in \mathbb{R}^{n \times k}$ represents the activation $A$ in terms of the dictionary atoms in $V$.

Different concept extraction methods approach this decomposition through various techniques. Some methods utilize traditional tools like K-means and PCA, others employ Non-negative Matrix Factorization (NMF), while Sparse Autoencoders (SAEs) have recently garnered significant attention due to their ability to be trained using Stochastic Gradient Descent (SGD) and their scalability to unseen data. We focus specifically on NMF and SAE in our discussion, in the dictionary learning framework from Fel et al. (2023), they can be unified as:

$$(U^*, V^*) = \arg\min_{U,V} ||A - UV^T||_F^2 \quad s.t. \begin{cases} U \geq 0, V \geq 0 & \text{(NMF)} \\ U = \psi(A), ||U||_0 \leq K & \text{(SAE)} \end{cases} \tag{1}$$

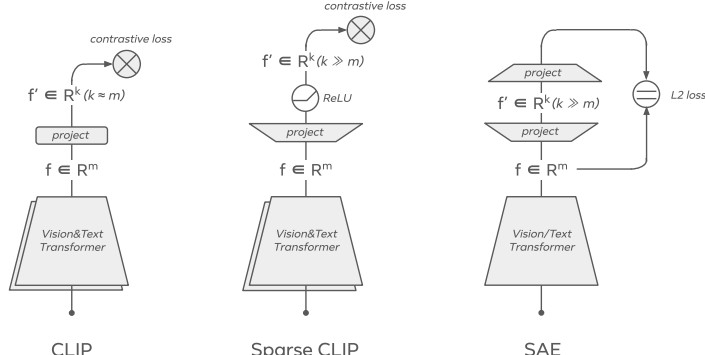

Figure 1: Comparison between Sparse CLIP and two existing methods, CLIP and SAE.

We initially explored adding sparsity to CLIP pretraining by incorporating traditional SAE components such as reconstruction loss and top-K activation into the CLIP training process. However, we quickly discovered that effective sparsity could be achieved with surprisingly minimal modifications to the original CLIP training procedure, requiring only two key changes:

1. **Non-negative constraints**: Adding ReLU activation after the final projection layer.

2. **Dimension expansion**: Significantly increasing the dimensionality of the final projection layer in CLIP.

Our findings turned out to be no coincidence. HaoChen et al. (2022) demonstrated that contrastive learning, when expressed in its spectral form, is equivalent to a Matrix Factorization (MF) objective. Building upon this foundation, Wang et al. (2024) proved that non-negative contrastive learning is equivalent to Non-negative Matrix Factorization (NMF), with this equivalence extending to the multimodal contrastive learning domain. Therefore, the theoretical foundations established in these papers provide justification for why our modifications—particularly the introduction of non-negative constraints—successfully induce sparsity in CLIP training. And putting this equivalence together with equation 1, we can start to view non-negative CLIP and SAE as different decomposition approaches under the same dictionary learning framework.

However, non-negativity constraints alone are insufficient for learning a satisfactory dictionary. Our small-scale experiments demonstrate that dimension expansion is critical for enabling sparse representations to achieve competitive performance on downstream tasks. This observation aligns with dictionary learning theory, where a sufficiently large dictionary size is required for effective representation learning. To the best of our knowledge, this work is the first to apply both non-negativity constraints and dimension expansion to CLIP training.

We demonstrate the comparison between CLIP training, sparse CLIP training and SAE in figure 1. After adding the high-dimensional-projection and non-negative constraint (ReLU) into vanilla CLIP, Sparse CLIP is close to SAE without decoder, but still trained with constrastive loss. In the next two sections, we will describe our training details and show benchmark results comparison of these three architectures.

## 2.2 PROOF OF CONCEPT ON SMALL SCALE TRAINING

We use OpenCLIP (Ilharco et al., 2021), a well-established open-source CLIP training repository, as our experimental foundation. To identify the optimal training recipe for sparse CLIP, we conduct small-scale ablation studies using a ViT-B/32 model and 15M image-text pairs sampled from the MetaCLIP dataset (Xu et al., 2024b). We evaluate performance using zero-shot classification accuracy on ImageNet-1k (Deng et al., 2009) and L0 sparsity metrics. Detailed experimental setup is provided in Appendix A.1. These small-scale experiments yielded several interesting observations:

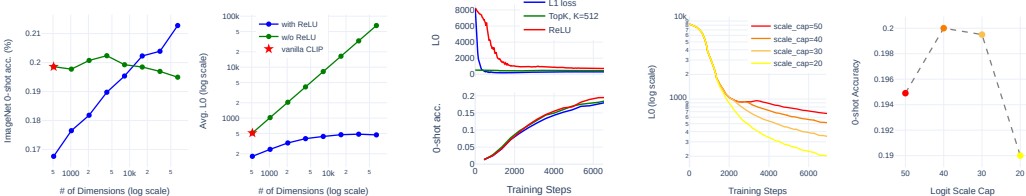

(a) Impact of non-negativity and embedding dimensionality (log scale) on model performance and sparsity.

(b) Comparison of sparsity injection methods.

(c) Impact of logit scale cap on sparsity and model performance. Logit scale (or 'temperature') is a learnable scaler for activations.

Figure 2: Small-scale ablation study. We evaluated different Sparse CLIP design choices by training ViT-B/32 models from scratch on a 15M sample dataset.

*Observation 1: In sparse CLIP training, both performance and sparsity increase with dimension scaling, but only under non-negativity constraints*. Figure 2a shows how non-negativity constraints and dimension expansion—our two primary CLIP training modifications—affect model performance and representation sparsity. The left panel reveals that increasing embedding dimensionality improves performance only when non-negativity constraints are enforced via ReLU activation. Without dimension expansion, non-negativity constraints alone degrade performance (leftmost point on blue curve). The right panel shows that activated features plateau as embedding dimensionality increases, but only under non-negativity constraints—without them, all features remain activated regardless of dimensionality.

*Observation 2: ReLU enables sparsity to form more gradually, improving model performance*. Figure 2b compares how different sparsity injection methods affect both sparsity levels and performance. We tested three sparsity-inducing techniques while keeping all other training parameters identical: L1 loss, TopK activation (K=512) Gao et al. (2024), and ReLU activation. Unlike typical SAE training, we apply these methods directly with CLIP loss rather than reconstruction loss. The top panel shows that L1 and TopK methods dramatically reduce L0 sparsity from training onset, potentially limiting model capacity and impairing learning effectiveness. The bottom panel confirms this hypothesis: ReLU achieves significantly superior zero-shot accuracy, demonstrating that gradual sparsity development preserves the model's learning capacity during training.

*Observation 3: Sparsity can be controlled by tuning the logit scale cap*. CLIP uses a learnable logit scale (temperature) parameter that multiplies cosine similarity scores before applying softmax/cross-entropy loss. This parameter controls softmax distribution sharpness, enabling the model to amplify differences between positive and negative pairs and learn from increasingly challenging examples during training. We discovered that sparsity levels can be effectively controlled by adjusting the logit scale cap. As shown in Figure 2c (left panel), reducing the logit scale cap consistently decreases L0 sparsity in activations. While we lack a complete theoretical explanation, this provides a practical sparsity control mechanism. However, optimal sparsity exists within a specific range. The right panel demonstrates this trade-off: when the logit scale cap drops to 20, zero-shot performance declines substantially, indicating that excessive sparsity reduction harms the model's representational capacity.

After conducting comprehensive ablation studies, we demonstrated the promising potential of sparse CLIP training on smaller models and datasets. These results motivated us to scale up our experiments with larger-scale training runs.

## 2.3 SCALING UP

Based on the findings on smaller models and datasets, we scale up both model size and data to train an interpretable CLIP model with performance comparable to mainstream models. With hyperparameter settings mostly consistent with our smaller-scale experiments, we use ViT-L/14 and train with the complete 2.2 billion sample MetaCLIP corpus for approximately 6 epochs. We apply a dimension expansion factor of 72[1], yielding a representation dimension of 55,296. Based on our observation

---

[1]An expansion factor of 72 is the maximum we can achieve for ViT-L/14 given 80GB GPU memory constraints. This limitation is discussed further in Section 5.

| Model | Zero-Shot Classification | | | | | | | Zero-Shot Fine-Grained Classification | | | | | | | | |
| --- | --- | --- | --- | --- | --- | --- | --- | --- | --- | --- | --- | --- | --- | --- | --- | --- |
| | Avg Class. | ImageNet | ImageNet-v2 | ObjectNet | ImageNet-A | ImageNet-R | ImageNet-S | Avg Fine. | Foods | Flowers | Pets | Cars | Aircrafts | Countries | Scenes | Satellite |
| OpenAI ViT-B/32 | **57.5** | **68.8** | **60.1** | **53.5** | **29.4** | **77.8** | **56.4** | **63.3** | **85.5** | **73.3** | **89.6** | **86.1** | **24.6** | **17.3** | **67.8** | 62.5 |
| **Prisma ViT-B/32** | 56.6 | 67.5 | 59.2 | 52.4 | 28.0 | 76.8 | 55.7 | 62.1 | 84.5 | 69.5 | 89.1 | 85.0 | 22.9 | 16.8 | 66.0 | **63.3** |
| ViT-L/14 baseline | 75.1 | **78.0** | **71.5** | 75.9 | 67.8 | 90.6 | **67.1** | 73.3 | 93.4 | **79.2** | 93.1 | 92.0 | 50.8 | 33.4 | **74.2** | 70.4 |
| **ViT-L/14 S. (0.66%)** | **75.6** | 77.1 | 70.2 | **77.2** | **71.9** | **91.0** | 66.5 | **74.0** | 94.0 | 79.0 | **93.1** | **92.0** | **53.7** | **34.7** | 74.0 | 72.0 |
| **ViT-L/14 S.+ (0.47%)** | 75.1 | 77.0 | 69.6 | 76.8 | 70.9 | 90.7 | 66.1 | 73.2 | **94.0** | 78.1 | 92.9 | 91.8 | 48.5 | 34.1 | 73.9 | **72.9** |

Table 1: Zero-shot image classification performance of Sparse CLIP models compared to their standard (non-sparse) CLIP counterparts.

| Model | Sparsity | BBox Classification | | Zero-Shot Retrieval | | | |
| --- | --- | --- | --- | --- | --- | --- | --- |
| | | Acc@1 | Acc@5 | IR@1 | IR@5 | TR@1 | TR@5 |
| ViT-L/14 OpenAI | 100% | 52.3 | 84.1 | 36.5 | 61.0 | 56.4 | 79.3 |
| ViT-L/14 baseline | 100% | 53.3 | 82.2 | **45.5** | **70.1** | **62.7** | **83.2** |
| **ViT-L/14 Sparse** | 0.66% | 55.5 | **86.1** | 43.7 | 68.5 | 59.9 | 83.0 |
| **ViT-L/14 Sparse+** | 0.47% | **56.0** | 85.3 | 41.8 | 66.4 | 57.0 | 80.5 |

Table 2: Results on additional CLIP downstream tasks. BBox classification simulates CLIP usage in open-vocabulary detection pipelines by classifying ground-truth bounding boxes. Zero-shot retrieval evaluates bidirectional image-caption matching, both on COCO validation set.

that logit scale cap affects sparsity, we train two models with logit scale caps of 50 and 40, producing models with different sparsity levels (0.66% vs. 0.47%). We name them ViT-L/14 Sparse and ViT-L/14 Sparse+. Following Bolya et al. (2025), we evaluate both models against the non-sparse baseline model on comprehensive zero-shot classification benchmarks (Table 1) and additional tasks (Table 2) to validate generalization ability.

*Sparse CLIP Performance*: As shown in Table 1, our ViT-L/14 Sparse model outperforms the dense baseline on both standard zero-shot classification tasks (+0.5% average across 6 benchmarks) and fine-grained classification tasks (+0.7% average across 6 benchmarks). The ViT-L/14 Sparse+ model performs slightly worse than ViT-L/14 Sparse but remains nearly on par with the baseline. On additional downstream tasks (Table 2), both ViT-L/14 Sparse and Sparse+ significantly outperform the baseline on bounding box classification, yet consistently underperform on zero-shot retrieval. We hypothesize that the combination of low L0 sparsity and the reduced logit scale cap encourages Sparse CLIP to focus predominantly on the dominant subject within each image. This behavior may be disadvantageous for COCO retrieval, where captions typically describe multiple subjects simultaneously.

*Comparison with SAE*: For comparison, we evaluate Prisma SAE (Joseph et al., 2025b) on the same zero-shot classification benchmarks (Table 1). Since Prisma SAE's sparse features were trained only on the vision branch, we evaluate using reconstructed dense features. Although trained solely on ImageNet, Prisma SAE generalizes well across domains but suffers 1-2% performance loss due to imperfect reconstruction. In contrast, our Sparse CLIP models match or surpass baseline performance while maintaining extreme sparsity.

# 3 MULTI-MODAL REPRESENTATIONS

Having demonstrated that Sparse CLIP models maintain competitive performance while achieving high sparsity, we now evaluate the interpretability enabled by their sparse representations.

## 3.1 INTERPRETABILITY EVALUATION

We first address an important question: How does Sparse CLIP interpretability compare to SAEs? We hypothesize that enforcing sparsity natively during training produces more interpretable features than post-hoc SAE approaches, and we demonstrate this through quantitative metrics.

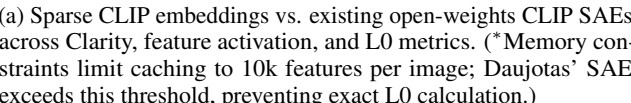
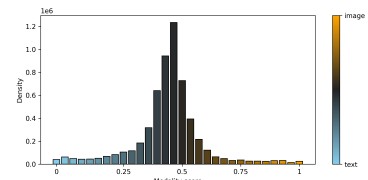

| Model | ImageNet-1k train (1.2M) | | | MetaCLIP subset (800k) | | |
|---|---|---|---|---|---|---|
| | Clarity ↑ | Active F% ↑ | L0 ↓ | Clarity ↑ | Active F% ↑ | L0 ↓ |
| dataset baseline | 0.485 | – | – | 0.422 | – | – |
| Prisma cls@11 | 0.519 | 45.1% | 916.0 | 0.509 | 54.8% | 1007.9 |
| Daujotas' SAE* | 0.521 | 41.0% | > 10k | 0.450 | 40.5% | > 10k |
| **ViT-L/14 Sparse** | 0.549 | **88.3%** | 468.5 | 0.542 | **89.1%** | 385.5 |
| **ViT-L/14 Sparse+** | **0.559** | 85.5% | **344.3** | **0.557** | 87.7% | **281.2** |

(a) Sparse CLIP embeddings vs. existing open-weights CLIP SAEs across Clarity, feature activation, and L0 metrics. (*Memory constraints limit caching to 10k features per image; Daujotas' SAE exceeds this threshold, preventing exact L0 calculation.)

(b) Distribution of Sparse CLIP features by modality score (image activation ratio) and activation density (magnitude-weighted frequency).

Figure 3: Quantitative interpretability evaluations of Sparse CLIP embeddings.

Standard SAE evaluation metrics such as Reconstruction Error and Fraction of Variance Unexplained (FVU) are not applicable to Sparse CLIP representations, as they lack a decoder for reconstruction. Instead, we employ the **Clarity** metric proposed by Dreyer et al. (2025), which measures the interpretability of visual embeddings in a manner agnostic to dimensionality, architecture, and training procedure.

Clarity quantifies feature interpretability by measuring the average pairwise cosine similarity between images that activate each feature, averaged across all active features. Formally, let $\mathcal{I}_i = \{x \in \mathcal{D} : a_i(x) > \tau\}$ denote the set of images activating feature $i$ above threshold $\tau$, let $\mathcal{F}_{\text{active}} = \{i : |\mathcal{I}_i| \geq n_{\min}\}$ be the set of features with at least $n_{\min}$ activating images, and let $\text{sim}(u, v) = \frac{u \cdot v}{\|u\|\|v\|}$ denote cosine similarity. The **Clarity** of visual embedding $e$ on dataset $\mathcal{D}$ is:

$$\text{Clarity} = \frac{1}{|\mathcal{F}_{\text{active}}|} \sum_{i \in \mathcal{F}_{\text{active}}} \frac{1}{|\mathcal{I}_i|(|\mathcal{I}_i| - 1)} \sum_{\substack{x_j, x_k \in \mathcal{I}_i \\ j \neq k}} \text{sim}(e(x_j), e(x_k)) \tag{2}$$

We evaluate Clarity on two datasets: the ImageNet-1k training set (∼1.2M images) and a subset of MetaCLIP containing 800k images. We employ the original OpenAI CLIP vision encoder as $e(x)$ and set $\tau = 0.001$ and $n_{\min} = 2$ to capture more activated features. For completeness, we also report the percentage of activated features and L0, since Clarity only measures activated features. Table 3a presents the results. Compared to open-weights CLIP SAEs (Joseph et al., 2025a; Daujotas, 2024), Sparse CLIP embeddings consistently outperform across all three metrics. While the ViT-L/14 Sparse+ model achieves a lower L0 at the cost of slightly reduced performance, it surpasses ViT-L/14 Sparse on Clarity, making it our primary focus for subsequent interpretability evaluations.

Next, we evaluate whether Sparse CLIP learns genuinely multimodal concepts—features that activate for both image and text inputs. Papadimitriou et al. (2025) previously trained SAEs on pretrained CLIP's multimodal representations but found most features were unimodal, activating exclusively for either images or text. Using the same evaluation methodology on ViT-L/14 Sparse+ features, we observe a dramatically different outcome. As shown in Figure 3b, Sparse CLIP representations are dominated by truly multimodal features. This stark contrast with post-hoc SAE approaches demonstrates a key advantage of native multimodal sparse training—the ability to learn genuinely cross-modal concepts.

## 3.2 Assigning Concept to Features

Interpreting vision features has long challenged researchers, typically requiring separate classification models or LLM assistance. Sparse CLIP offers a direct solution: we can name features using vocabulary words that trigger maximum activations. If our features are truly multimodal, these word-based labels should accurately reflect the visual concepts each feature encodes.

To validate this approach, we constructed a comprehensive vocabulary with minimal conceptual overlap. We selected the first word from each WordNet synonym set (McCrae et al., 2019) and merged these with the 20K most common English words (Oikarinen & Weng, 2023) which covers informal ones from Internet. This combination ensures broad coverage of both formal and colloquial English, yielding a final vocabulary of 98,619 words.

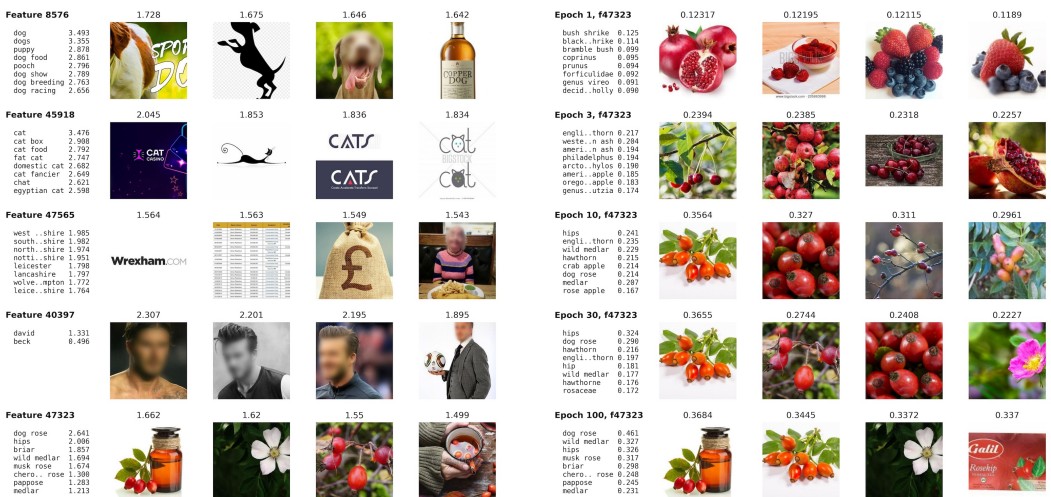

Figure 4: Examples features of ViT-L/14 Sparse+. The top activated words and images for these multimodal features are highly correlated, so we can directly name their visual concept based on text.

Figure 5: Evolution of "dog rose" feature. Top activated words and images at different stage of training. Activations values are normalized.

We analyzed each ViT-L/14 Sparse+ feature by identifying the top-K words and images that produced the highest activations, using this 98,619-word vocabulary and 80K randomly sampled MetaCLIP images. We found high correlations between text and visual concepts for most features, indicating that Sparse CLIP multimodal features indeed activate on semantically similar content across both modalities. Figure 4 shows example features with their top-activated words and images. The "dog" and "cat" features at the first two rows demonstrate robust visual understanding, with top images spanning realistic photos, abstract logos, and OCR text. The "British" feature captures cultural symbols including currency, buildings, and food, while its text activations consist primarily of city names—a pattern observed in many geographical concepts. Additional examples include an exclusive "David Beckham" feature and a mixed "dog rose/rose hips" feature. More interesting example features can be found in Appendix A.2.

### 3.3 CONCEPT EMERGENCE AND EVOLUTION

The last three features in Figure 4 seem to suggest that there may be patterns in how multimodal concepts emerge. Do they begin as separate single-modal features that merge later? Also, will their learnt concepts evolve during training?

While many studies have proposed methods to interpret what concepts CLIP models have learned, few have investigated how concepts emerge and evolve during training. This gap is understandable given the fact that intermediate training checkpoints are typically unavailable for open-source models. However, we found training Sparse CLIP from scratch with native interpretability provides a unique opportunity to examine concept emergence and evolution in an easy-to-visual multimodal setting.

We analyzed ViT-L/14 Sparse+ checkpoints at 1%, 3%, 10%, 30%, and 100% training completion, visualizing top-activated words and images for each feature at every stage. This reveals feature lifecycles—how concepts emerge, develop, and mature throughout training. Our analysis provides insights into not only what CLIP learns, but how it acquires knowledge from large-scale image-text data through contrastive learning.

*Observation 1: Concepts emerge as multimodal features early in training.* Figure 6 shows the modality distribution at 1% training completion, which closely resembles the final distribution—most features activate for both images and text from very early stages. The primary difference is activation density: due to lower early-training sparsity, densities are an order of magnitude higher than at completion. Examining specific features at 1% completion (Figure 7) confirms strong correlation

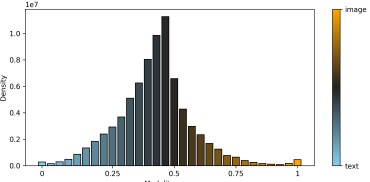
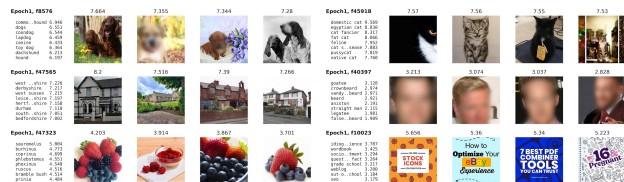

Figure 6: Modality distribution of Sparse CLIP features similar to Figure 3b, but at 1% training completion.

Figure 7: Same example features as figure 4, but at 1% training completion, which is after the first training epoch.

| Vision Encoder | MMMU | AI2D | TextVQA | POPE |
|---|---|---|---|---|
| ViT-L/14 baseline | 39.6 | 67.2 | 48.9 | 80.8 |
| **ViT-L/14 Sparse** | 40.8 | 66.4 | **51.3** | **82.0** |
| **ViT-L/14 Sparse+** | **41.7** | **70.6** | 48.5 | 80.7 |

Table 3: Image QA benchmark results. We train VLMs with different CLIP vision encoders, and compare the VLM performance on image QA benchmarks that covers different areas.

between image and text activations. However, some early concepts differ substantially from their final forms, leading to our second observation.

*Observation 2: Feature concepts evolve and sometimes completely transform during training.* Analyzing top activation trajectories uncovers how concepts evolve during training. Consider the "dog rose" feature (Figure 5): initially activating for red fruits and random creatures, it narrows to rose hips mid-training before crystallizing into a dog rose detector that retains hip activations—clear evidence of emergent multimodal alignment. We observe similarly striking transformations: feature 40397 progresses "goatee" → "Ryan Gosling" → "David Beckham"; feature 10884 shifts from eye makeup to conjunctivitis; feature 38195 evolves from generic soccer players to "Lionel Messi" (Appendix A.2). These patterns raise fundamental questions about whether transformations are knowledge-driven or noise-driven, their correlation with performance, and their controllability—opening promising avenues for future work.

## 4 APPLICATIONS

CLIP's widespread adoption makes interpretable multimodal representations valuable for applications like controllable image generation, open-vocabulary detection, and image search. To demonstrate our approach's viability, we focus on Vision-Language Models (VLMs) due to their prominence and fundamental reliance on CLIP's cross-modal capabilities.

We trained a minimal VLM using ViT-L/14 Sparse+ as the vision encoder and Llama 3.1 8B Instruct as the language model, connected via a 2-layer MLP adapter. Our VLM training followed a staged approach: first pretraining the adapter with frozen vision encoder (Sparse CLIP models) and the LLM (LLama 3.1 8B instruct), then fine-tuning the adapter and LLM together. First stages uses SA-1B-1M (Kirillov et al., 2023) for 2 epoch and second stage used 2.5M image-QA pairs from PLM image dataset (Cho et al., 2025) for 4 epochs. Table 3 shows our sparse VLM achieves comparable performance across four benchmarks covering perception, diagrams, hallucination, and general image-based QA.

Our primary demonstration focuses on vision-based steering (Joseph et al., 2025a). As illustrated in Figure 8a, manipulating sparse representation activations prior to the adapter layer enables precise output control—transforming "dog" to "cat" concepts in generated text, or suppressing sensitive concepts like "password" for security applications.

We systematically evaluate steering effectiveness on the complete ImageNet-1k validation set (50k images) using the prompt: "What's the main subject in this image? Your answer needs to be short". Our steering protocol for each image involves two operations: (1) suppressing the ground truth label by zeroing its top-K activated features in the ViT-L/14 Sparse+ text encoder, while (2) boosting a

|  | Example 1 | Example 2 |
|---|---|---|
| **Input** | (image) | (image) Password generator |
| **Output before steering** | "The image depicts a young dog, likely a puppy, ... The dog has a fluffy ..." | "Password generator" |
| **Steering** | feature 8576 (dog): $1.2129 \rightarrow 0.0$ feature 45918 (cat): $0.0 \rightarrow 2.0$ | feature 16340 (password): $1.3618 \rightarrow 0.0$ |
| **Output after steering** | "The image features a *cat* sitting on a table... The *cat* is wearing a collar ... The *cat's* fur is ..." | "A blue box with a white window in the middle and a white input box below the window." |

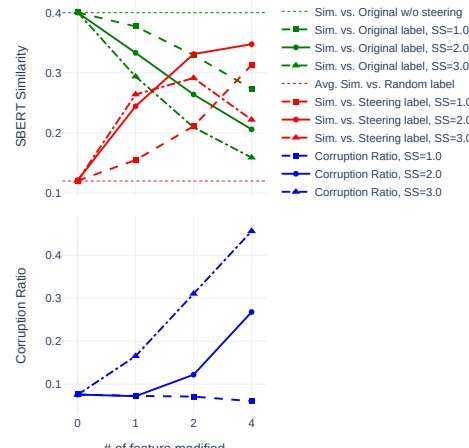

(a) Qualitative examples of concept steering. We query the VLM with "What's in the picture?" while applying steering (steering strength is 2.0 for boosting, 0.0 for suppression) to the features most strongly activated by target concept words.

(b) Steering evaluation on ImageNet-1k. We measure steering effectiveness (SBERT cosine similarity) toward random ImageNet labels across different numbers of modified features and Steering Strengths (SS).

Figure 8: Concept steering in vision-language models, with ViT-L/14 Sparse+ as the vision encoder.

randomly selected alternative ImageNet-1k label by setting its top-K activations to Steering Strength (SS). Figure 8b presents performance across different K and SS configurations, revealing the following observations:

*Suppressing top-activated concept features effectively removes concepts from outputs.* Measured by SBERT cosine similarity, VLM outputs move away from the ground truth label after steering (green lines in Figure 8b top plot). This effect scales linearly with both the number of modified features and the steering strength of the new concept.

*Boosting top-activated features of new concepts effectively steers outputs toward them, but risks model corruption.* Similarly, VLM outputs move toward the boosted alternative label after steering (red lines in Figure 8b top plot). However, this effect is moderated by model corruption, shown in the bottom plot of Figure 8b and measured by identifying abnormally long responses. At high SS values, corruption risk increases sharply with the number of modified features, causing outputs to diverge in unintended directions. The results suggest that SS = 2.0 provides effective steering while minimizing corruption when modifying only 1–2 features, which we adopt for the examples shown in Figure 8a.

## 5  LIMITATIONS AND FUTURE WORK

**Training for native interpretability**: Unlike most mechanistic interpretability research that analyzes existing models, we trained CLIP variants to be natively interpretable. This fundamental difference enables comparable performance while gaining interpretability and revealing concept evolution during training. While insights from our sparse models don't directly transfer to existing dense CLIP models, comparing learned concepts across architectures presents compelling future work.

**Parameter overhead**: Interpretability incurs computational costs—our projection layers require significantly more parameters to map representations to higher-dimensional sparse space. In ViT-L/14, the linear projection from 768 to 768×72 dimensions adds 14% of parameters to the vision tower. Whether performance gains stem from added parameters versus sparsity remains an open question for future investigation.

**Memory constraints**: CLIP training requires gathering batch activations across GPUs for cosine similarity computation. GPU memory becomes a bottleneck for sparse CLIP due to representation scaling. Despite extensive optimization, 72× expansion is the max we can achieve on 80GB GPUs.

Optimizing sparse CLIP training for memory efficiency and throughput remains an important research challenge.

# 6 RELATED WORKS

**Improving CLIP training.** Since CLIP was first proposed (Radford et al., 2021; Jia et al., 2021), there has been extensive research aimed at improving its training. Most efforts focus on enhancing the downstream performance of representations by refining the loss functions (Zhai et al., 2023; Li et al., 2023; Fini et al., 2024) or improving the quality of training data (Xu et al., 2024b;a; Schuhmann et al., 2022). Recently, Perception Encoder (Bolya et al., 2025) provided a comprehensive summary of these advancements and established a new state-of-the-art in performance. In our work, we follow the benchmarks used in Perception Encoder to evaluate our model.

**Sparse Autoencoders for CLIP.** Sparse Autoencoders (SAEs) were originally developed to analyze LLMs and have since been applied to interpret CLIP models (Lim et al., 2025; Zaigrajew et al., 2025). Most notably, Joseph et al. (2025b) trained SAEs on each layer of the CLIP vision transformer, compared vision models with LLMs, and systematically evaluated CLIP SAE steering capabilities (Joseph et al., 2025a). Their publicly released SAE weights serve as our primary comparison baseline. More recently, Papadimitriou et al. (2025) trained SAEs on the multimodal representation space of pretrained CLIP models and found that most features were unimodal. In contrast, our analysis demonstrates that features learned through sparse CLIP training are predominantly multimodal: they are activated by both visual and textual inputs and represent similar concepts across modalities.

**Sparse CLIP Representations.** Wang et al. (2024) showed that introducing non-negativity into contrastive learning induces sparsity, as it is equivalent to Non-negative Matrix Factorization (NMF). Although they did not explore increasing the dimensionality to strengthen the representations, their work provides a theoretical foundation for our approach. In a different direction, Chen et al. (2023) constructed sparse CLIP representations by projecting dense features into a high-dimensional space predefined by a vocabulary. While this approach yields interpretable and multimodal representations, the predefined space limits the model's ability to capture high-level semantics beyond the vocabulary, thus restricting its capacity to fully leverage the richness of the training data.

**Interpretability and Performance.** A common view is that interpretability comes at the cost of accuracy: sparsity and disentanglement can improve feature clarity but reduce reconstruction or predictive power (Olshausen & Field, 1996; Higgins et al., 2017; Koh et al., 2020), and recent studies show that sparsity constraints often hurt training stability or downstream accuracy (He et al., 2022; Xie et al., 2024; Nasibullah et al., 2024; Chen et al., 2025; Sawmya et al., 2025). This belief motivates the dominance of post-hoc interpretability methods such as Sparse Autoencoders (Elhage et al., 2022; Cunningham et al., 2023), which are applied to the model after it is trained. In contrast, other work demonstrates that interpretability and performance can be co-optimized: interpretable models can rival black-box accuracy (Rudin, 2019), concept embedding models and modular networks maintain strong performance while improving transparency (Zarlenga et al., 2022; Swamy et al., 2023; Good et al., 2023), and prototype-based vision models even improve accuracy alongside interpretability (Zhu et al., 2025). Our work extends this latter line to multimodal learning, showing that sparsity imposed during CLIP training yields interpretable features without accuracy loss.

# 7 CONCLUSION

We introduced Sparse CLIP, which enforces sparsity in CLIP training to produce interpretable multimodal features without sacrificing performance. Unlike post-hoc sparse methods such as Sparse Autoencoders, which often reduce accuracy and collapse multimodal structure, our approach preserves performance, and we demonstrate its utility through vision-language steering. This challenges the longstanding view that interpretability and accuracy are in tension. If confirmed across architectures and modalities, it suggests a paradigm shift: interpretability and performance need not be trade-offs, but can be jointly optimized as core design principles.

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

## A  APPENDIX

### A.1  SMALL-SCALE ABLATION STUDY SETUP

For Sparse CLIP small-scale ablation studies, we follows OpenCLIP original setup for most of the hyperparameters, except for the following.

**Training Dataset**: We uses MetaCLIP as the source of our training data. It's a large-scale dataset containing around 2.2 billion image-text pairs collected from the Internet and indexed by Common-Crawl, then curated following the methodologies described in Xu et al. (2024b) and Xu et al. (2024a). For our preliminary experiments, we utilize a randomly sampled subset of MetaCLIP containing approximately 15 million samples, and train the model for 30 epochs on this dataset.

**Model Architecture**: For our preliminary experiments, we employ the default ViT-B/32 configuration from OpenCLIP to enable fast training while maintaining reasonable model capacity. This architecture uses a 12-layer Vision Transformer (ViT) for both the vision and text encoders, with an embedding dimension of 512.

**Input Image Processing**: Input images are resized to 224×224 pixels for efficient training. Non-square images are resized based on their shortest edge and then center-cropped to preserve aspect ratios.

**Loss Function**: We apply the standard CLIP loss, which computes the average cosine similarity loss on both visual and text embeddings during training.

**Representation Dimensionality**: We use a 32× dimension expansion (16,384 dimensions for ViT-B/32) unless otherwise specified in ablation studies.

**Batch size and Learning rate**: Following OpenCLIP's configuration, we use a learning rate of 5e-4 and global batch size of 32K, implemented via either 512×64 or 128×256 (batch size per GPU x GPUs).

**Evaluation Metrics**: We use zero-shot classification accuracy on the ImageNet-1K (Deng et al., 2009) test set as the primary metric for evaluating model performance during training. For measuring sparsity, we use average L0 value of the activation.

### A.2  ADDITIONAL QUALITATIVE RESULTS

#### A.2.1  FEATURE VISUALIZATION

As mentioned in Section 3.2, we show visualization for more interesting example features here in Fig 9.

#### A.2.2  FEATURE EVOLUTION

Many features exhibit fascinating evolution trajectories during training. We provide detailed visualizations for the representative examples mentioned in Section 3.3.

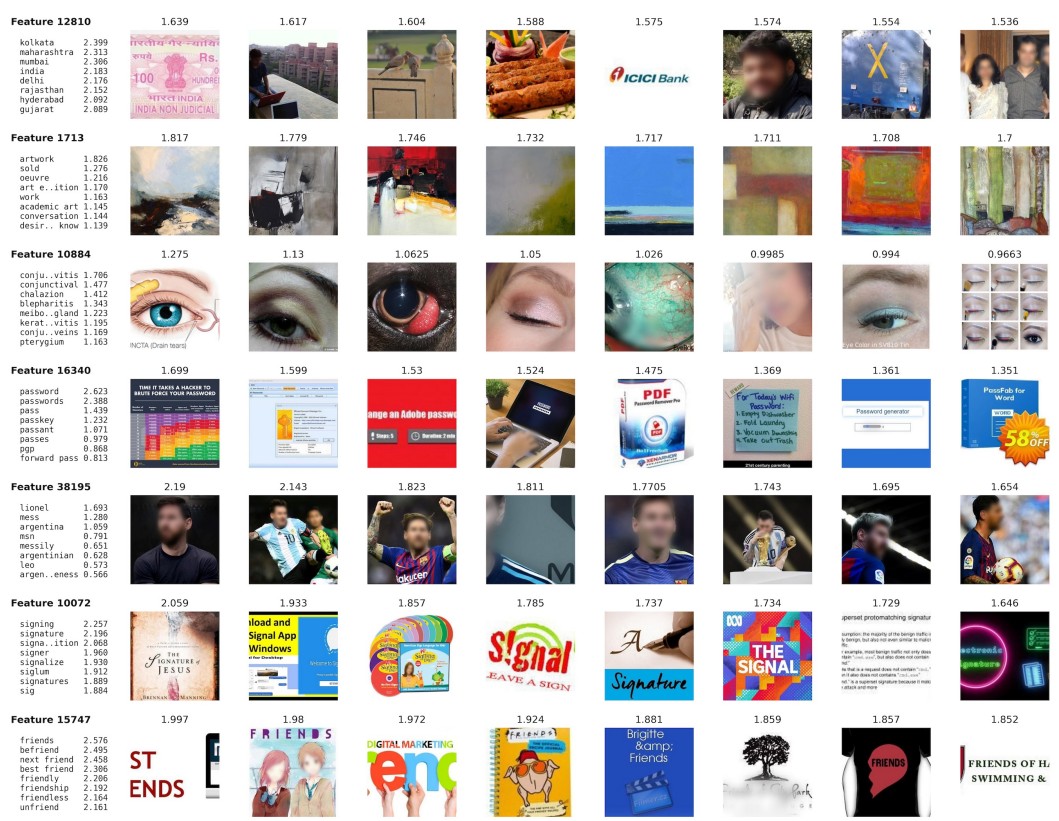

Figure 9: More example ViT-L/14 Sparse+ features with their highest activated words and images. Notice that we show un-normalized activation values here.

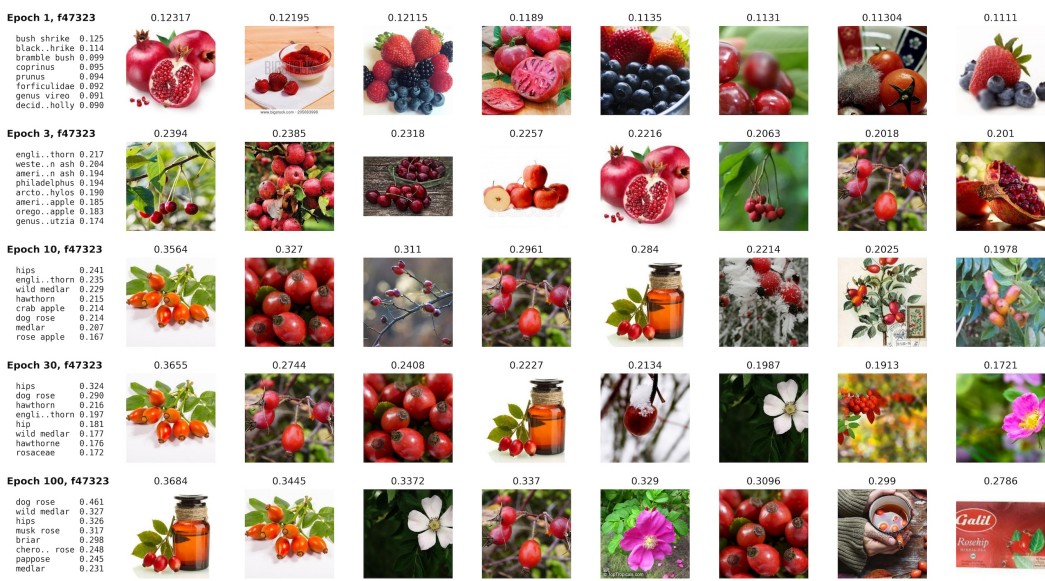

Figure 10: A full picture for evolution of the "dog rose" feature.

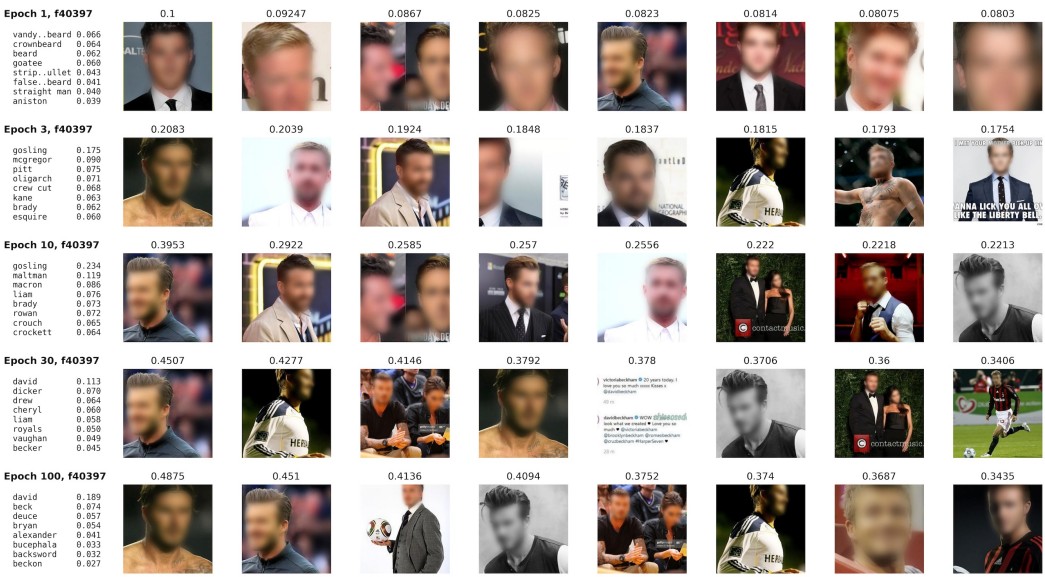

Figure 11: Evolution of the "David Beckham" feature. It starts as a feature for "beard", turned into "gosling" in the middle of training, and finally become a dedicated "David Beckham" feature.

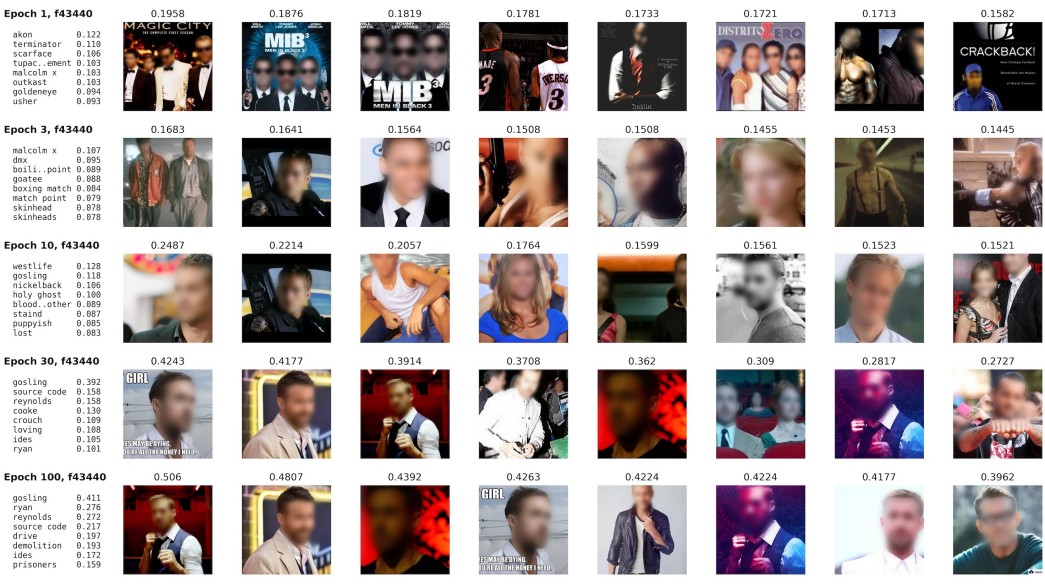

Figure 12: Evolution of the "Ryan Gosling" feature. It become Gosling at the same time when the original Gosling feature taken by David Beckham, as shown in Figure 11.

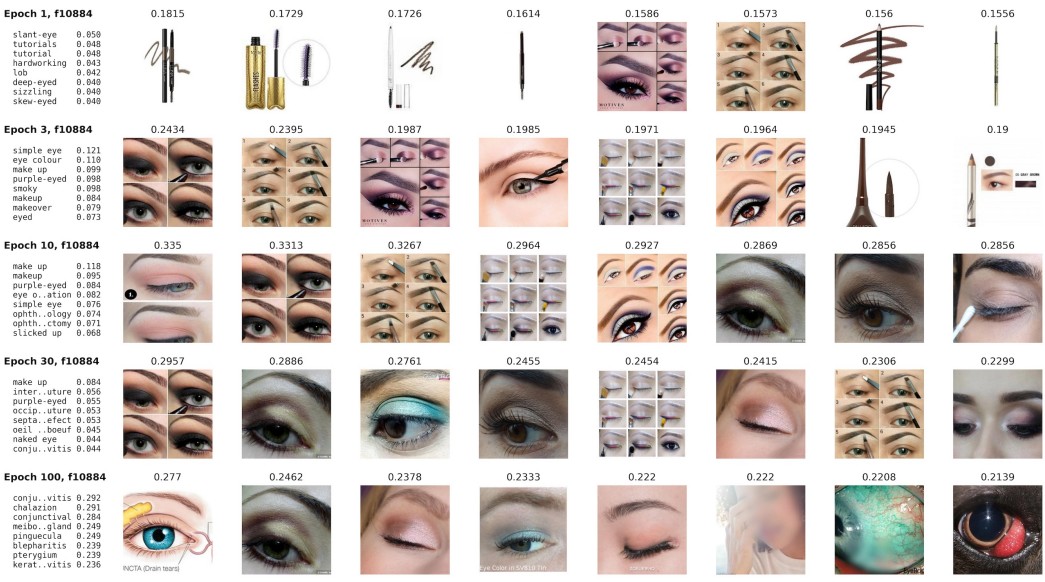

Figure 13: Evolution of the "conjunctivitis" feature: Initially, this feature activates for eye makeup products but eventually converges to "conjunctivitis" in text while still activating for some eye makeup images. This suggests the model may struggle to distinguish between eye makeup and inflamed eyes.

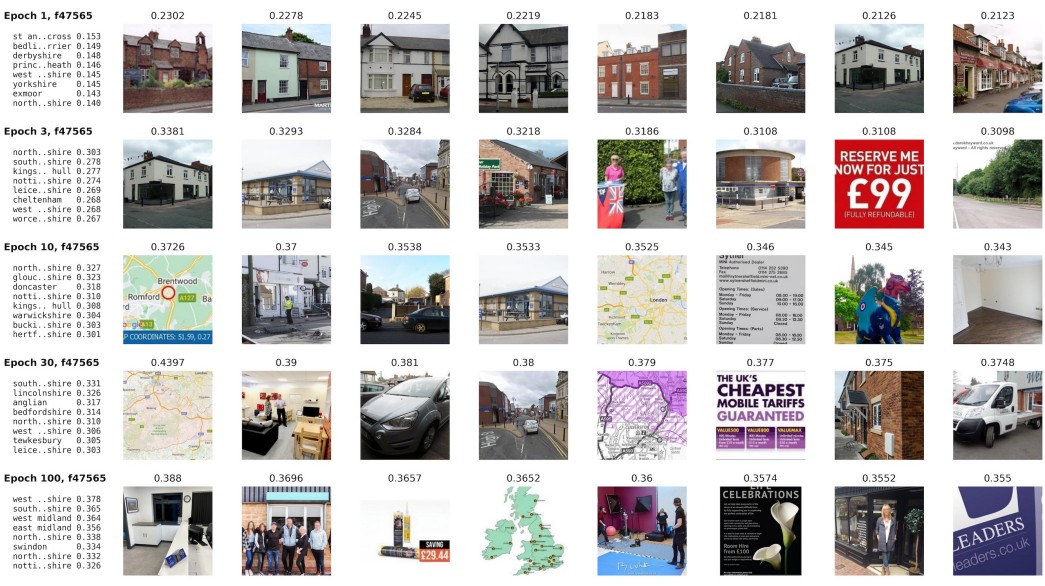

Figure 14: Evolution of the "British" feature. The feature begins by activating for images of British-style houses during early training, then gradually generalizes to represent a wider range of geographical and cultural concepts.

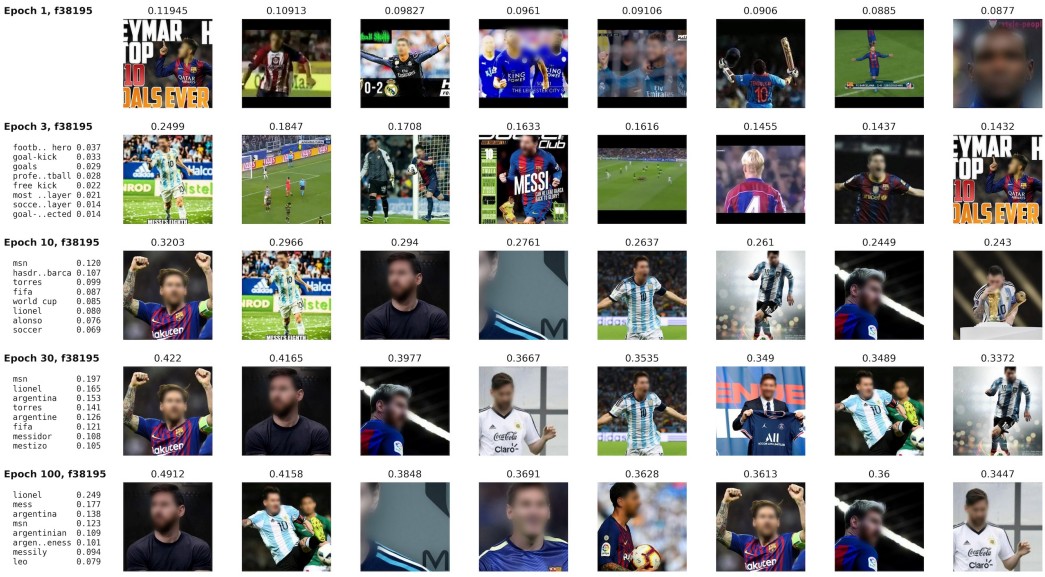

Figure 15: Evolution of the "Messi" feature. It starts as an image-only feature for a group of sports pictures, turned into a soccer only feature in the middle of training, and finally became the dedicated "Messi" feature on both modalities.

