

Figure 1: Example with heatmap using ViT-L/14 Sparse-Avg. We overlay the spatial token activations for each feature as heatmaps on the corresponding images, using the Sparse CLIP model trained with average-pooled spatial tokens instead of CLS token.

# SUPPLEMENTARY MATERIAL FOR SPARSE CLIP: CO-OPTIMIZING INTERPRETABILITY AND PERFORMANCE IN CONTRASTIVE LEARNING

**Anonymous authors**

## 1 CONCEPT ACTIVATION ON SPATIAL TOKENS

In CLIP training, the contrastive loss is typically applied to the CLS token, which is generally considered to contain a compressed representation of all spatial tokens and thus captures the semantic information of the image. In Sparse CLIP training, we inherited this setup and computed the loss on the CLS token. However, when we attempted to visualize spatial localization using the learned concepts from the CLS token, the results did not produce meaningful heatmaps—the sparse activations on the CLS token do not directly transfer to spatial tokens.

Instead of using the CLS token, we computed the contrastive loss using average-pooled activations across all spatial tokens, while keeping all other hyperparameters identical to ViT-L/14 Sparse. This resulted in a model with comparable performance and L0 sparsity, which we refer to as ViT-L/14 Sparse-Avg.

We found that concept heatmaps work significantly better with ViT-L/14 Sparse-Avg. Figure 1 shows representative examples: similar to the visualization in the main paper, we show several features with

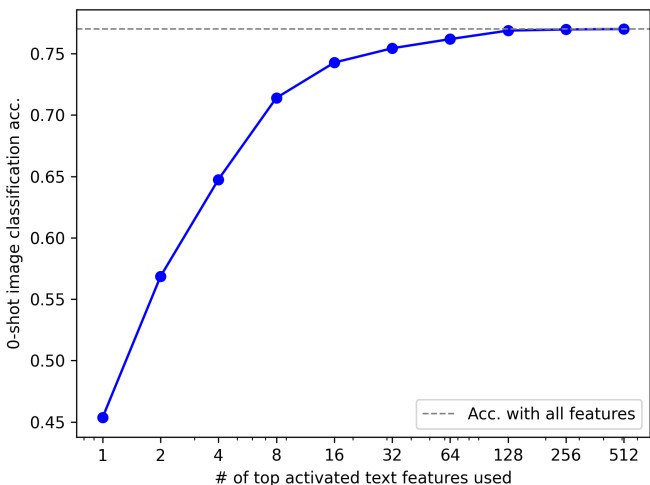

Figure 2: Zero-shot ImageNet-1k classification using top activated features of ViT-L/14 Sparse+ on class labels.

their top-activated words and images, but now we also overlay the spatial token activations for each feature as heatmaps on the corresponding images.

The heatmaps show strong alignment with the subject matter (indicated by the highest-activated word) across diverse visual representations—including portraits, cartoons, silhouettes, text, and even artistic fonts. However, we also observed that **the heatmaps lack spatial precision**: high activations sometimes occur at spatial tokens distant from the actual subject.

## 2 ZERO-SHOT IMAGENET-1K CLASSIFICATION WITH SPARSE FEATURES

As shown in Figure 2, we performed zero-shot ImageNet-1k classification using the top activated features of ViT-L/14 Sparse+ on ImageNet-1k labels. We found that:

- Using only the top 1 activated feature achieves 45% accuracy
- Using the top 32 features nearly matches the performance obtained with all features

These results demonstrate that a small subset of highly activated sparse features captures most of the discriminative information.

## 3 C-SCORE FOR TEXT EMBEDDINGS

We extend the C-score metric from the main paper to measure the interpretability of text embeddings. Specifically, we compute the feature-wise C-score as the average pairwise cosine similarity of all words (from a given vocabulary) that activate a feature, then average across all features to measure the interpretability of a text embedding. We use SBERT features instead of CLIP features on text since it gives more diverse results and more efficient to get. Since we lack other text embeddings for comparison (existing open-weight CLIP SAEs are only trained on vision encoders), we evaluate ViT-L/14 Sparse+ across different activation thresholds. The results are shown in Figure 3.

On the right side of the figure, we observe that when restricting to activations larger than 1.5, the C-score reaches 0.45, substantially higher than the average pair-wise SBERT similarity baseline of 0.16 on our vocabulary. Given that the average L0 is 1.26 in this setting, this provides a quantitative measure of interpretation quality when using the top-activated word to label a feature's concept.

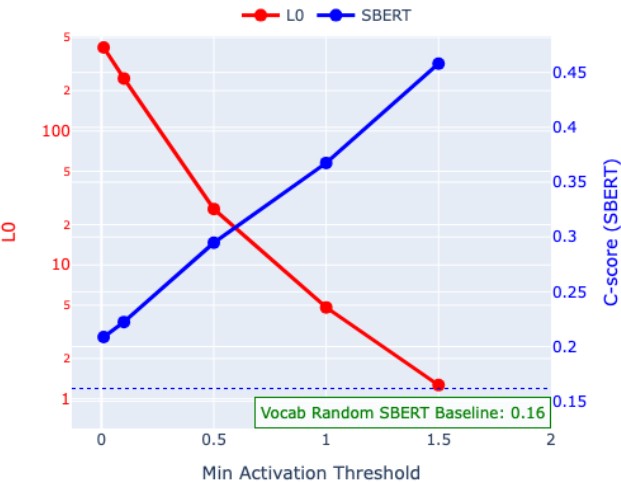

Figure 3: C-score for text embeddings on our 10k vocabulary, with different thresholds applied on minimal activations.