# OpenReview forum: "Sparse CLIP: Co-Optimizing Interpretability and Performance in Contrastive Learning"
_ICLR.cc/2026/Conference — ICLR 2026 Poster_

### Official Review · Reviewer_6eqm · 2025-10-27

**Soundness:** 3
**Presentation:** 3
**Contribution:** 3
**Rating:** 6
**Confidence:** 4

**Summary:**

This paper challenges the prevailing assumption that interpretability must come at the cost of performance in representation learning. The authors propose Sparse CLIP, a method that seamlessly co-optimizes for both objectives by integrating sparsity directly into the Contrastive Language-Image Pre-training (CLIP) framework. The approach is notably simple, requiring only two key modifications to the standard architecture: a significant expansion of the final projection layer's dimensionality and the introduction of a ReLU activation function. This combination effectively induces a sparse, high-dimensional representation space without altering the core contrastive learning objective.

Empirical results demonstrate that Sparse CLIP not only matches but in some cases surpasses the performance of its dense counterparts on zero-shot classification benchmarks, while achieving extreme sparsity. Crucially, the learned features are natively multimodal, activating for semantically aligned concepts across both image and text modalities. This enables direct concept-level interpretability, allowing features to be named and analyzed based on their top-activating words and images. The paper further reveals intriguing training dynamics, tracing how multimodal concepts emerge and evolve, and showcases the practical utility of these interpretable features through vision-based steering in a downstream Vision-Language Model (VLM), enabling controlled generation and security filtering.

**Strengths:**

1. Simplicity and Effectiveness: The method simply increases the dimensionality of the final projection layer and adds an activation of ReLU. The result shows that it achieves interpretability and better performance.
2. Refutation of a prevailing assumption: It provides a good counterexample for "training-time sparsity compromises accuracy".
3. High application potential: The paper demonstrates performance enhancements in downstream applications, as well as new application scenarios such as steered generation and security filtering.

**Weaknesses:**

1. Theoretical innovation is limited: One of the key techniques (ReLU activation) was proposed by NCL. It lacks a theoretical analysis of dimensionality expansion. The emphasis is more on applications in multimodal setting.
2. GPU memory and model parameters overhead: The model significantly increases the dimensionality of the final projection layer, resulting in substantially more parameters and the additional GPU memory required for training. Due to limitations of the GPU memory, it is impossible to further research the impact of dimensionality.
3. Limited evolution for downstream task: Only four fine-tuning tasks are reported, which is insufficient to robustly support the claim that the method "maintains strong performance on downstream tasks" across a broad range of applications.

**Questions:**

1. In 2.3, the distinction between the training settings for Sparse and Sparse+ is not clarified. Do they merely employ different logit scale caps? If so, based on the conclusions from small scale training, is the Sparse+ model trained using a logit scale cap of 40?
2. Model Scale: Since the final projection layer is significantly enlarged, could the performance gain be attributed primarily to the increase in parameters rather than the sparsity mechanism? It is necessary to supplement the Sparse CLIP model with a set of ablation experiments—specifically, removing the ReLU activation function and dimension expansion individually.
3. Association between text and features: The paper explains that one should first establish a vocabulary list, then identify the corresponding relationships. This demonstrates dependence on vocabulary quality for concept labeling.
    1. Can the model interpret multi-word phrases, or is it limited to single tokens?
    2. How is it verified that the top-activated word accurately captures the feature's semantic role?
    3. What happens if the underlying concept is not contained in the vocabulary?
4. Steering Sensitivity: In the "dog"→"cat" steering example, the "cat" feature is set to 2.0.
    1. How was this value chosen? Would other values (e.g., 1.0 or 10.0) lead to unpredictable generation?
    2. Is there a systematic method for determining safe and effective intervention magnitudes?

---

> ### Author Response · Authors · 2025-11-24
>
> We sincerely appreciate Reviewer 6eqm's thoughtful and constructive comments and suggestions.
>
> We address most questions and concerns raised, specifically:
>
> - Limited evaluation for downstream task
> - Difference between Sparse and Sparse+
> - Ablation for ReLU and dimension expansion
> - Association between text and image
> - Steering sensitivity
>
> Details below:
>
> ## 1. Limited Evaluation for Downstream Tasks
>
> We conducted additional experiments on two different downstream tasks.
>
> **Bounding Box Classification:** This task simulates CLIP usage in open-vocabulary detection pipelines by classifying ground-truth bounding boxes from the COCO validation set (36,781 bbox over 5k images). **Sparse CLIP significantly outperforms the dense baseline.**
>
> **Zero-Shot Retrieval:** Bidirectional image-caption matching on COCO validation set (IR/TR metrics). **Sparse CLIP underperforms the baseline.** We hypothesize that the combination of low L0 and the smaller logit scale cap encourages Sparse CLIP to focus more on the dominant subject in each image, whereas COCO captions often describe multiple subjects. This limitation warrants future investigation.
>
> Results added to Section 2.3. While these findings largely support our claim that "Sparse CLIP preserves downstream task performance," they also illuminate important directions for future improvements.
>
>
> ## 2. Difference Between ViT-L/14 Sparse and Sparse+
>
> Only difference: logit scale cap value (50 vs. 40). This was clarified in the revised paper—original had a LaTeX typo omitting the explanation sentence.
>
>
> ## 3. Ablation for ReLU and Dimension Expansion
>
> We'd like to note that these ablations are already presented in Figure 2(a), which showing performance changes from 1x to 128x dimension expansion with/without ReLU:
>
> - **Dimension expansion requires non-negativity:** Performance improves only with ReLU
> - **Non-negativity alone insufficient:** Without expansion, non-negativity degrades performance
>
> Both components work synergistically—neither effective alone.
>
> **On parameter count concern:** We acknowledge this concern and have explicitly addressed it as a limitation in Section 5 of the original paper. However, additional parameters scale with sparse dimension, not model size. 72× expansion adds ~14% parameters for ViT-L/14, only ~3% for ViT-G/14. Moreover, Figure 2(a) shows expansion without non-negativity (green line) fails, indicating sparsity mechanism itself is essential, not just parameter count. We agree this question warrants deeper investigation. Fully disentangling the contributions of parameter count versus sparsity mechanisms represents a valuable direction for future research. In the current work, we acknowledge the increased parameter count as a necessary tradeoff for achieving interpretability.
>
>
> ## 4. Association Between Text and Image
>
> **Multi-word phrases?** Yes. Vocabulary based on WordNet includes 30.22% multi-word phrases.
>
> **Verification of top-activated words?**
> - **Qualitative:** Visualized all features with highest-activating images and words, showing strong semantic alignment (examples in Section 3.2 and in Appendix A.2)
> - **Quantitative:** Figure 2 in supplement shows zero-shot ImageNet-1k classification using the top-activated features from ViT-L/14 Sparse+ on ImageNet-1k labels: top-1 achieves 45% accuracy; top 32-64 nearly matches all features. It demonstrate that the top-activated features for a given ImageNet label strongly align with the visual content of the corresponding images.
>
> Another more direct quantitative metrics is described in the answer for the next question.
>
> **Missing vocabulary concepts?** It could happen. Ultimately, can we evaluate interpretability on a vocabulary? In Figure 3 of supplement, we extend C-score (new metrics we added in Section 3 of the updated paper) to text embeddings: average pairwise cosine similarity of words activating each feature. With activations >1.5, C-score (SBERT) reaches 0.45 vs. baseline 0.16 (average L0=1.26), quantifying interpretation quality when using top-activated words.
>
>
> ## 5. Steering Sensitivity
>
> We added quantitative evaluation for VLM steering in Section 4 to investigate steering sensitivity.
>
> **Benchmark:** ImageNet-1k test set (50k images, 1k classes). VLM with ViT-L/14 Sparse+ queries: "What's the main subject in this image? Your answer needs to be short."
>
> **Protocol:**
> - Suppress ground truth by zeroing top-K features
> - Boost alternative label by setting top-K to Steering Strength (SS)
>
> **Results (K ∈ [0,1,2,4], SS ∈ [1.0,2.0,3.0]):**
> - **Suppression works:** Outputs shift away from ground truth, scaling linearly with K and SS
> - **Boosting works but risks corruption:** Outputs shift toward target, but high SS sharply increases corruption. SS=2.0 with 1-2 features provides effective steering with minimal corruption
>
>
> ---
>
> We hope these clarifications and additional results resolve all questions and outstanding concerns. We are happy to answer any remaining questions.

---

### Official Review · Reviewer_sRWz · 2025-10-29

**Soundness:** 3
**Presentation:** 2
**Contribution:** 3
**Rating:** 6
**Confidence:** 4

**Summary:**

This work addresses the interpretability limitations of CLIP’s dense multimodal representations. The authors propose integrating sparsity directly into CLIP training to create “Sparse CLIP features” that remain highly performant on downstream tasks while becoming significantly more interpretable. Unlike post-hoc sparse autoencoder methods, this approach preserves CLIP’s inherent multimodal alignment and reveals transparent feature evolution and concept emergence during training. Experiments further demonstrate practical benefits by enabling interpretable visual-language control in downstream applications. The results challenge the common belief that interpretability must come at the cost of performance, offering a promising direction for building multimodal models that are both accurate and understandable.

**Strengths:**

1. This paper proposes the SPARSE CLIP method, which effectively combines the strengths of both CLIP and SAE models, ensuring interpretability and performance in contrastive learning.
2. The paper provides a solid analysis of the interpretability achieved by combining large models with SPARSE CLIP. The cases presented in paper demonstrate the strong interpretability of SPARSE CLIP as well as its powerful application potential.

**Weaknesses:**

The proposed SPARSE CLIP method exhibits strong interpretability, but its performance drops on many benchmarks compared to the original ViT-based CLIP baseline. I hypothesize that this is due to CLIP’s reliance on maintaining rich and continuous directional information in the embedding space. The discontinuity introduced by ReLU indeed produces highly sparse and interpretable features, but it also collapses many feature dimensions, resulting in sparse vectors that weaken cross-modal alignment and semantic expressiveness. A potential solution would be to decouple the interpretable sparse features from the output by employing an independent sparse decoder.

**Questions:**

The paper presents many interpretability examples and analyzes the distribution of Sparse CLIP features, yet it lacks more fine-grained analysis that connects specific feature activations to corresponding image regions or pixels. For instance, applying masking-based methods could enable more detailed ablations to identify which parts of the image activate particular features. I hope the authors can include some cases demonstrating this aspect.

If the authors provide reasonable and satisfactory responses to the Weaknesses and Questions, I would consider increasing the score.

---

> ### Author Response · Authors · 2025-11-24
>
> We truly thank sRWz for the nice comments and helpful suggestions.
>
> We address all questions and concerns raised, specifically:
>
> - About the concern for discontinuity
> - About feature activation per region
>
> Below, we provide detailed responses to each of them:
>
> ---
>
> ## 1. About the Concern for Discontinuity
>
> The review raised an great point: adding ReLU could collapse many feature dimensions, potentially weakening cross-modal alignment and semantic expressiveness. This is a reasonable hypothesis. Our argument is that despite ReLU collapsing some feature dimensions, the 72x dimension expansion provides substantially more capacity, allowing the model to learn effectively from training data while maintaining sparsity.
>
> This concern is closely related to another question we have not fully explored in the paper: **What is the optimal trade-off between interpretability and performance?** We presented two Sparse CLIP models with different sparsity levels (based on our finding that L0 can be controlled via logit scale capping). We observed that the sparser model sacrifices some performance for better interpretability (we added quantitative interpretability comparisons in Section 3.1 of the updated paper). Whether there exists an optimal balance point remains an open question.
>
> We also conducted an experiment directly related to the reviewer's concern. We tested replacing ReLU with a hybrid ReLU + GeLU approach, where ReLU is applied during the forward pass and GeLU during backpropagation. This technique allows gradients to flow through small negative activations. We found that it reduces dead features but increases L0, with final performance metrics comparable to standard ReLU. Since dead features are not a major issue in Sparse CLIP training (<10%), we chose ReLU for simplicity.
>
> ---
>
> ## 2. About Feature Activation Per Region
>
> In CLIP training, the contrastive loss is typically applied to the CLS token, which is generally considered to contain a compressed representation of all spatial tokens and thus captures the semantic information of the image. In Sparse CLIP training, we inherited this setup and computed the loss on the CLS token. However, when we attempted to visualize spatial localization using the learned concepts from the CLS token, the results did not produce meaningful heatmaps—the sparse activations on the CLS token do not directly transfer to spatial tokens.
>
> To address this limitation, we explored an alternative training configuration. Instead of using the CLS token, we computed the contrastive loss using average-pooled activations across all spatial tokens, while keeping all other hyperparameters identical to ViT-L/14 Sparse. This resulted in a model with comparable performance and L0 sparsity, which we refer to as **ViT-L/14 Sparse-Avg**.
>
> We found that concept heatmaps work significantly better with ViT-L/14 Sparse-Avg. We've shown representative examples in Figure 1 of the updated supplement with the paper. In the Figure we show several features with their top-activated words and images, but now we also overlay the spatial token activations for each feature as heatmaps on the corresponding images.
>
> The heatmaps show strong alignment with the subject matter (indicated by the highest-activated word) across diverse visual representations—including portraits, cartoons, silhouettes, text, and even artistic fonts. However, we also observed that **the heatmaps lack spatial precision**: high activations sometimes occur at spatial tokens distant from the actual subject.
>
> We were unable to resolve this issue before the submission deadline and ultimately decided to remove these results from the submitted paper. Moving forward, we plan to revisit this direction. Other recent work has successfully trained SAEs on spatial tokens, and we believe Sparse CLIP has strong potential to achieve competitive or superior spatial interpretability with further refinement.
>
> ---
>
> We hope these clarifications and additional results resolve all questions and outstanding concerns. We are happy to answer any remaining questions. Thank you for your detailed and thoughtful review.

---

### Official Review · Reviewer_BAoM · 2025-10-30

**Soundness:** 2
**Presentation:** 1
**Contribution:** 4
**Rating:** 4
**Confidence:** 4

**Summary:**

This paper proposes training a vision-language encoder (CLIP) with a drastically expanded embedding dimension (in this case, by a factor of 72). The goal is to learn a sparse representation allowing for interpretability and steering. Experimental results show that Sparse CLIP is able to perform well in zero-shot classification on ImageNet, as well as its representation can be interpreted through concept discovery.

**Strengths:**

This paper's strengths lie in the novelty and simplicity of the method:
1. The idea to train a CLIP with an inherently sparse representation is very interesting and can lead to significant progress in research on interpretability. It is simple and could be easily translated to other bi-modal encoders trained with the contrastive loss.
2. I appreciate the paper's flow, especially the 'lessons learned' given in Section 2.2, articulating the research process progressing into Section 2.3.

**Weaknesses:**

Overall, this work is in an early stage, requiring revision and extension to be considered for publication:
1. Regarding soundness, the experiments are limited (see questions below). Contribution 3 (implementing interpretable
vision-based steering in vision-language models) executed in Section 4 is overstated significantly: the two examples shown in Table 2 cannot reliably demonstrate real-world applicability.
2. The paper requires proofreading (see feedback below; I just stopped listing at some point). Several elements in the paper appear to be missing (see questions), and I do not believe that mentioning additional content exists in the Appendix and then incorporating it during the rebuttal would be an acceptable practice.
3. Furthermore, the comparison with the state-of-the-art is missing. Section 3 could follow experimental evaluation from the omitted related work on training SAEs for CLIP [a,b] instead of comparing to a TopK SAE with a single $k$ value (not to mention that the Prisma SAE uses a different baseline model).
4. The paper should include code to reproduce the results.

[a] Sparse autoencoders reveal selective remapping of visual concepts during adaptation. ICLR 2025

[b] Interpreting CLIP with hierarchical sparse autoencoders. ICML 2025

**Questions:**

1. The sentence in L200-201 "After conducting comprehensive ablation studies (additional results in Appendix A.3)," is wrong. There are no "comprehensive ablation studies" in Appendix A.3. It only shows Figure 9 with a single experiment on "ViT-L/14 Sparse+".
2. Appendix A.2 mentions an "interactive web visualization to access visualizations" but the demo is empty, i.e. says "Cool Sparse CLIP visualizations".
3. Why did you choose the dimension expansion factor of 72? Comprehensive ablations are essential when proposing a method with such a critical parameter. Why does the Appendix say "32x dimension expansion"?
4. Table 1 should include results for ViT-B/32 Sparse.

**Other feedback**:
- Use `\citep` correctly instead of `\citet` (many such cases e.g. in L153-157).
- L137: add "equation" to "1"
- L149: typo in "architectures"
- Figure 1:
    - Be more specific in the caption, e.g. add the information about "OpenCLIP (ViT-B/32) trained on MetaCLIP".
    - Improve colors in (c); use an actual gradient, e.g. yellow-orange-red-purple.
    - Graphs are pixelated; improve their quality, e.g. export them to PDF.
- Figure 9 (Appendix A.3): It is completely unclear what "ViT-L/14 Sparse+" or this result means when linking here from L200.
- L206: "After demonstrating proof of concept on smaller models and datasets," reads like an overstatement, because it was a single model (ViT-B/32) and data subset.
- L212: typo in "(0.66Following"
- Table 1/L232: Again, it is unclear what does "+" in "ViT-L/14 Sparse+" mean.
- L244: Again, "Having achieved satisfactory performance and sparsity" reads like an overstatement, because the model was trained with a single hyperparameter setting and evaluated only on the zero-shot classification task. We know little regarding the generalizability of the approach.
- Table 3 (Appendix A.1.2) should be a part of main text.
- L420: What does "Not on existing model" mean?

---

> ### Author Response · Authors · 2025-11-24
>
> We sincerely appreciate the thoughtful and constructive comments. We have addressed the three major concerns with substantial revisions:
>
> - Limited experimental results for VLM steering (Section 4)
> - Comparison with state-of-the-art CLIP SAEs
> - Generalizability of the approach
>
> All proofreading issues have been corrected. Below are detailed responses:
>
> ---
>
> ## 1. Limited Experimental Results for VLM Steering
>
> We appreciate the suggestion to further investigate VLM steering. We acknowledge that the original Section 4 lacked quantitative evaluation and steering methodology.
>
> We added quantitative evaluation for VLM steering.
>
> **Benchmark:** We built a steering benchmark using ImageNet-1k test set (50k images, 1k classes). VLMs (ViT-L/14 Sparse+ vision encoder) query each image with prompt: "What's the main subject in this image? Your answer needs to be short."
>
> **Steering Protocol:**
> - Suppress ground truth label by zeroing its top-K activated features
> - Boost random alternative label by setting its top-K activations to Steering Strength (SS)
>
> **Results (K ∈ [0,1,2,4], SS ∈ [1.0,2.0,3.0]):**
> - **Suppression works effectively:** SBERT similarity moves away from ground truth, scaling linearly with modified features and SS
> - **Boosting works but risks corruption:** Outputs shift toward target label, but high SS increases corruption sharply. SS=2.0 with 1-2 features provides effective steering with minimal corruption
>
> We hope this extra experiment result can strengthen the VLM steering case. All results are plotted and added to Section 4.
>
> ---
>
> ## 2. Comparison with State-of-the-Art CLIP SAEs
> We acknowledge the concern regarding the lack of quantitative comparison of interpretability with state-of-the-art CLIP SAEs.
>
> **Challenges:** Standard SAE metrics (Reconstruction Error, FVU) are inapplicable since Sparse CLIP lacks a decoder. VLM-as-judge approaches are computationally prohibitive.
>
> **Solution:** We propose **Concept Score (C-score)**, measuring interpretability via average pairwise cosine similarity among images activating each feature. Details in revised Section 3.1.
>
> **Results:** We compared against two open-weights CLIP SAEs (Prisma SAE and Daujotas' SAE) on ImageNet-1k (1.2M images) and MetaCLIP (800k images). **Sparse CLIP consistently outperforms both across C-score, activation percentage, and L0 metrics.**
>
> We hope this new quantitative result helps understanding how Sparse CLIP embeddings compared to SOTA CLIP SAEs, and we’ve rewrite most of section 3.1 with these new results.
>
> ---
>
> ## 3. Generalizability of the Approach
>
> Reviewer pointed out two types of generalizability concerns:
> - Generalizability on different training hyperparameters
> - Performance generalizability on more downstream tasks
>
> ### 3.1 Training Hyperparameters
>
> We tried different training hyperparameters in early experiments, and found most of them did not significantly affect the final results. Here is a summary of other options we’ve investigated:
>
> - **Model size:** Trained ViT-B/16 on 2.2B dataset—improvements align with ViT-L/14
> - **Activations:** ReLU forward + GeLU backward reduces dead features but increases L0; TopK variants show no improvement
> - **Spatial Token pooling:** Average pooling comparable to CLS token
> - **Fine-tuning:** No benefits over training from scratch
> - **SigLIP:** Prohibitively slow training
>
>
> ### 3.2 Downstream Tasks
> We conducted additional experiments on new downstream tasks.
>
> **Bounding Box Classification:** This task simulates CLIP usage in open-vocabulary detection pipelines by classifying ground-truth bounding boxes from the COCO validation set (36,781 bbox over 5k images). **Sparse CLIP significantly outperforms the dense baseline.**
>
> **Zero-Shot Retrieval:** Bidirectional image-caption matching on COCO validation set. **Sparse CLIP underperforms the baseline.** We hypothesize that the combination of low L0 and the smaller logit scale cap encourages Sparse CLIP to focus more on the dominant subject in each image, whereas COCO captions often describe multiple subjects.
>
> We've added these results to Section 2.3.
>
> ---
>
> ## Additional Questions
>
> **Why 72x dimension expansion?** Figure 2(a) shows performance scales linearly with dimensionality, and 72x is maximum achievable on 80G GPU memory with ViT-L/14 and our batch size. Appendix 1.1's "32x" refers to small-scale ablation default—wording clarified.
>
> **Table 1 ViT-B/32 Sparse?** We didn't train large-scale ViT-B/32 Sparse. Table 1 compares Prisma vs. baseline to show SAE performance degradation limitation. Our contribution is demonstrating Sparse CLIP avoids this—comparison with dense baseline validates this claim.
>
> **Demo and code?** Institutional policies prevent our original plan on releasing demos/code/weights. We apologize for this limitation.
>
> ---
>
> We hope these clarifications and additional results resolve all questions and outstanding concerns. We are happy to answer any remaining questions. Thank you for your detailed and thoughtful review.

---

> > ### Comment · Reviewer_BAoM · 2025-11-26
> >
> > I appreciate authors who put in this significant effort.
> >
> > I have increased my rating accordingly.
> >
> > The above-mentioned references [a, b] should be cited, as they are closely related works.

---

### Official Review · Reviewer_SVSv · 2025-11-01

**Soundness:** 3
**Presentation:** 3
**Contribution:** 2
**Rating:** 4
**Confidence:** 3

**Summary:**

This paper introduces Sparse CLIP, a experimental modification and integration to the standard CLIP framework that incorporates sparsity during training.

**Strengths:**

1. The paper incorporates sparsity into training, it achieves both interpretability and high performance, offering a compelling design.

2. Comprehensive experiments demonstrate that Sparse CLIP performs comparably to dense CLIP on zero-shot classification tasks and surpasses post-hoc Sparse Autoencoders (SAEs) in interpretability.

3. Applications in Vision-Language Models: This paper presents a practical demonstration of Sparse CLIP’s utility and relevance for downstream tasks.

In summary, this paper offers intriguing experimental results and new insights into sparse training than providing a clear technical contribution or a robust methodological advancement.

**Weaknesses:**

1. While the findings are interesting, the proposed method is totally established on the existing works, making it relatively simple and lacks significant novelty.

2. Unclear generalizability: While Sparse CLIP performs well on zero-shot classification benchmarks, it is unclear how well the method generalizes to downstream tasks as the other CLIP models, such as object detection, segmentation, or open-vocabulary retrieval

**Questions:**

1. Introduce quantitative interpretability metrics beyond classification metrics.

2. Hard case testing and failure case analysis. For example, does Sparse CLIP produce meaningful features although it misclassifies an object? What is the performance of Sparse CLIP on more complex datasets?

3. Clarify contributions.

---

> ### Author Response · Authors · 2025-11-24
>
> We truly thank SVSv for the nice comments and helpful suggestions.
>
> We address all questions and concerns raised, specifically:
>
> - Quantitative interpretability metrics
> - Generalizability on different downstream tasks
> - Failure cases analysis
> - Clarification for our contributions
>
> Below, we provide detailed responses to each of them:
>
>
> ## 1. Quantitative Interpretability Metrics
>
> We acknowledge the concern regarding the lack of quantitative comparison of interpretability with state-of-the-art CLIP SAEs. This has been a significant challenge throughout the preparation of this paper, as establishing appropriate metrics for such comparison is non-trivial:
>
> **Challenges:** Standard SAE metrics (Reconstruction Error, FVU) are inapplicable since Sparse CLIP lacks a decoder. VLM-as-judge approaches are computationally prohibitive.
>
> **Solution:** We propose **Concept Score (C-score)**, measuring interpretability via average pairwise cosine similarity among images activating each feature. Details in revised Section 3.1.
>
> **Results:** We compared against two public CLIP SAEs (Prisma SAE and Daujotas' SAE) on ImageNet-1k (1.2M images) and MetaCLIP (800k images). **Sparse CLIP consistently outperforms both across C-score, activation percentage, and L0 metrics.** ViT-L/14 Sparse+ achieves superior C-score, justifying its use in interpretability evaluations.
>
> | Model | C-score↑ (IN/MC) | Active%↑ (IN/MC) | L0↓ (IN/MC) |
> |---|---|---|---|
> | Baseline | 0.485/0.422 | —/— | —/— |
> | Prisma | 0.519/0.509 | 45.1/54.8 | 916/1008 |
> | Daujotas | 0.521/0.450 | 41.0/40.5 | >10k/>10k |
> | Sparse | 0.549/0.542 | 88.3/89.1 | 469/386 |
> | Sparse+ | **0.559/0.557** | 85.5/87.7 | **344/281** |
>
>
> ## 2. Generalizability on Different Downstream Tasks
>
> We conducted additional experiments on two downstream tasks to evaluate model generalizability, as presented in the table.
>
> **Bounding Box Classification:** This task simulates CLIP usage in open-vocabulary detection pipelines by classifying ground-truth bounding boxes from the COCO validation set (36,781 bbox over 5k images). **Sparse CLIP significantly outperforms the dense baseline.**
>
> **Zero-Shot Retrieval:** Bidirectional image-caption matching on COCO validation set (IR/TR metrics). **Sparse CLIP underperforms the baseline.** We hypothesize that the combination of low L0 and the smaller logit scale cap encourages Sparse CLIP to focus more on the dominant subject in each image, whereas COCO captions often describe multiple subjects. This limitation warrants future investigation.
>
> Results added to Section 2.3. While these findings largely support our claim that "Sparse CLIP preserves downstream task performance," they also illuminate important directions for future improvements.
>
>
> ## 3. Failure Cases Analysis
>
> While we haven't conducted a systematic failure analysis, we did observe some interesting failure patterns during our investigations.
>
> **Granularity of Learned Concepts:** During examining ImageNet failures, we found fine-grained labels underperform generic ones. Example: "tench" (#0) has a lower success rate than label #1 "goldfish", and we found "tench" lacks a dedicated feature while "goldfish" has two dedicated features. Similarly, "tiger shark" (#4) underperforms "great white shark" (#3). We hypothesize limited feature dimensionality and training data representation prevent fine-grained concepts from developing dedicated features. They may be represented through feature composition though—an interesting future direction.
>
> **Data Bias Artifacts:** Training noise led features to capture meaningless patterns. For example, we found a dedicated feature for the term "undefined" (likely because it's a common uninitialized value in JavaScript), and another feature specifically for image watermarks from a particular website. This highlights needs for better dataset curation and suggests our method could potentially serve as a data cleaning tool.
>
>
> ## 4. Clarification for Our Contributions
>
> **On Simplicity as Strength:** While our method is simple (non-negativity constraints + dimension expansion with CLIP training), the implications are profound. **We challenge the prevailing assumption that interpretability requires sacrificing performance,** demonstrating this tradeoff is unnecessary. Simplicity is a strength—it suggests interpretability barriers were overestimated and enables extension to other domains.
>
> **On Multimodal Contribution:** Another major contribution is **the first multimodal-aligned interpretable representations in the literature** (to our knowledge), while outperforming CLIP SAEs in interpretability. We demonstrate unique capabilities multimodal alignment enables (cross-modal steering, analysis), representing significant potential for future interpretability research.
>
> ---
>
> We hope these clarifications and additional results resolve all questions and outstanding concerns. We are happy to answer any remaining questions. Thank you for your detailed and thoughtful review.

---

> > ### Comment · Reviewer_SVSv · 2025-11-26
> > **Reply to authors**
> >
> > Thank you to the authors for their responses to my questions. My concerns have been basically resolved. However, I recommend that the authors incorporate the interpretability metrics and generalization experiments into the main text to strengthen the paper. In addition, compared with the baseline CLIP SAEs, more comprehensive quantitative or qualitative feature-level analyses could be considered and included in the paper, to enhance the statement about "interpretable representations".
> >
> > I am willing to raise my score to a 6. Nevertheless, although the method is simple and effective, due to the lack of significant and essential technical innovation, I cannot raise the score to 8 currently.

---

### Meta-Review · Area_Chair_zZJD · 2026-01-06

**Summary:**

This paper initially received two positive scores and two negative score (Reviewer SVSv, BAoM) with an average score of 5. The main concerns lie in (1) Quantitative interpretability metrics (Reviewer SVSv, BAoM), (2) Downstream task generalizability (Reviewer SVSv, BAoM, 6eqm), (3) VLM steering evaluation (Reviewer BAoM, 6eqm), (4) Technical contributions and clarification (Reviewer SVSv, 6eqm), (5) Comparison with SOTA methods (Reviewer BAoM), (6) Fine-grained analysis of feature activations (Reviewer sRWz).
After the rebuttal, Reviewer SVSv indicated that most of his/her concerns are basically addressed, and the score is increased from 4 to 6. Reviewer BAoM indicated that he/she would increase the score. Meta review read the paper, reviews, rebuttal, and the author's message, which confirmed that the most concerns from both Reviewers SVSv and BAoM are well addressed, including quantitative interpretability, downstream task generalizability and technical contributions, etc. For the concerns on the discontinuity problem from Reviewer sRWz and VLM steering evaluation from Reviewer 6eqm, the authors provided the corresponding experiments and analysis, which is suitable to address these concerns.

Overall, the proposed method is simple, yet effective to achieve the interpretability-performance tradeoff by projecting dense features into a high-dimensional space and enforcing sparsity during CLIP training. The contributions are solid with comprehensive experiments. Thus, Meta reviewer recommends accepting this paper. Despite that, this paper lacks of the theory innovation as commented by Reviewer 6eqm, such that some doubts regarding its true interpretability still exist. Moreover, the performance comparison (rather than the interpretability) with more SOTA methods should be added to further evaluate the effectiveness of the proposed method.

**Reviewer Concerns:**

Most concerns are well addressed.

**Reviewer Scores:**

I think Reviewer SVSv and Reviewer BAoM would increase the score if they participated fully in the discussion.

---

### Decision · Program_Chairs · 2026-01-26

Accept (Poster)